# Application of Nanotechnology in Extinguishing Agents

**DOI:** 10.3390/ma15248876

**Published:** 2022-12-12

**Authors:** Anna Rabajczyk, Maria Zielecka, Justyna Gniazdowska

**Affiliations:** Scientific and Research Center for Fire Protection—National Research Institute, Nadwiślańska 213, 05-420 Józefów, Poland

**Keywords:** extinguishing agents, nanotechnology, application

## Abstract

Extinguishing agents are a very important tool in the field of security, both in terms of private and social aspects. Depending on the type of burning substance and place of fire, appropriately prepared and developed solutions should be used. We can distinguish, among others, materials, powders or foaming agents. Modifications introduced into them, including ones based on the achievements in the field of nanotechnology, can improve their safety of use and extend their service life. Such amendments also reduce the costs of production and neutralization of the area after a fire, and increase the fire extinguishing effectiveness. The introduction of nanoparticles allows, e.g., shortening of the fire extinguishing time, reduction of the risk of smoke emission and the toxic substances contained in it, and an increase in the specific surface of particles and thus increasing the sorption of pollutants. The elaborations use metal nanoparticles, e.g., NP-Ag, metal oxides such as NP-SiO_2_, as well as particles of substances already present in extinguishing agents but treated and reduced to nanosize. It should be noted, however, that all changes must lead to obtaining a tool that meets the relevant legal requirements and has appropriate approvals.

## 1. Introduction—Extinguishing Agents

A fire is defined as a chemical combustion process at which high-temperature oxidation of a combustible material (or fuel) takes place [1]. Oxygen, temperature, fuel and chemical reactions are required for the fire to catch and spread [2]. A fire can occur both inside a building and outdoors, and it can be at one point or may occupy a large area. It can include natural, artificial, modified, simple and complex substances. The problem of fires, especially in relation to large areas, is expanding from year to year as a consequence of climate change and the increasingly frequent high temperatures and hydrological droughts. Tidey [3] showed that there was a fourfold increase in the number of forest fires, as there were over 1900 fires between January and July 2022, while the average for 2006–2021 for this period is equal to 520. He also indicated that 2021 was the worst season in the EU since 2000 in terms of the number of forest fires and the area covered by the fire. It should be noted, however, that fires also occur in construction facilities, industrial plants, waste depots, material heaps or other facilities, which determines the number of fires in general. Over the last five years (2017–2021), just in Poland the number of fires amounted to an average of 132,813 [4], causing a number of socio-economic losses, i.e., losses in infrastructure, property, environment and above all losses in people.

The selection of an extinguishing agent is, therefore, a necessary element not only in respect to the prevention system, but also in respect to the subsequent fire extinguishing actions. The extinguishing agent must be adapted to the potential fire hazards, as it allows for adequate protection and minimization of the fire consequences. The choice of an extinguishing agent is also subject to the relevant regulations and constitutes a legal requirement, the purpose of which is to protect the place of the incident against the spread of fire and to maximally reduce the time of fire extinguishing. This obligation applies not only to firefighters during the firefighting action, but also to the owners and administrators of real property. It should be remembered that the combustion reaction releases energy, various substances and heat. Therefore, when extinguishing fires, it is important to select an appropriate extinguishing agent as well as extinguishing method and technique. This is defined by a number of parameters characteristic to each specific case, in particular by the fire class, and by the properties of substances and materials involved in the fire (especially by their physical state). A special position among fires is occupied by those involving vegetable and animal fats (class F of fire) [5,6]. Class A fires include fires of solids of organic origin, during which the phenomenon of incandescence occurs. Class B fires are fires of flammable liquids and substances that melt due to the heat generated by a fire, while class C fires include fires of gases, e.g., methane, acetylene, etc. Class D fire is a fire of metals, including magnesium, uranium and sodium [5,6]. Fires caused by electrical equipment are marked with the symbol of an electric spark, not the letter E [5,6]. It should be noted, however, that in some countries, such as the UK, there are fire classes such as:
–Class A—the presence of solids, including materials such as paper, wood and plastic.–Class B—the presence of liquids, such as paraffin, petrol and oil.–Class C—flammable gases, such as propane, butane and methane. –Class D—the presence of metal products, such as aluminum, magnesium and titanium. –Class E—caused by electricity or ones that involve electrical equipment and apparatus. –Class F—most commonly occur in kitchens and food preparation facilities, and involve cooking oil or fat [7]. 

Due to the variety of substances, elements that are affected by the fire, and the scale and possibilities of the fire spreading, the fire suppression mechanism of extinguishing agents also has to be diverse. Its effect can be obtained by removing heat from the surface of the burning material, cooling, cutting off the access of oxygen as well as by inhibiting radical reactions taking place in the flame. For this reason, there are numerous extinguishing agents (Figure 1) based on water, loose materials and gases.

Taking into account the above, extinguishing agents are subject to modifications and continuous improvement in order to better protect people, property and infrastructure in the event of a fire. Modifications introduced in recent years are increasingly based on nanotechnology (Figure 2). On the one hand, new substances are used, the sizes of which are in the nanoscale. On the other hand, scientists take advantage of the already known compounds, the particle sizes of which are getting smaller, which allows obtaining new properties of the same compounds. 

The introduced innovative solutions allow for the faster and more precise action of extinguishing agents. Nanoparticles are characterized by a larger adsorption surface, which results in faster reactions with compounds emitted during a fire. Nanoparticles often have the ability to produce reactive oxygen particles, which allows the degradation of organic, toxic compounds into simpler, less toxic or neutral compounds. The use of nanoparticles can therefore reduce the risk of smoke spreading along with the toxic substances contained in it, minimizing the threat to human life and health. 

Therefore, continuous research in this area is essential. It allows for the development of tools adequate to the changing materials, structures and substances that surround us. Tracking the effects of the introduced innovations also allows for determining further directions of the introduced solutions and constant improvement of tools in the area of fire protection. 

The aim of this review is to present the most important information on nanotechnology-based innovations in extinguishing agents, based on a literature review using the following keywords: extinguishing agents, fire extinguisher, foam, fire blanket, nanoparticles, fire protection, fire, smoke. The literature review was carried out using the following databases: Web of Knowledge, Scopus and Google Scholar. This study also covered Espacenet, Patentscope and Google Patents, which resulted in the presentation of selected patents related to the subject. In this paper, the analysis covers both literature reports and patent databases, reports of companies whose activities are related to fire protection. The information obtained will allow determination of the directions of introduced changes as well as their legitimacy in the context of effective fire protection. It should be noted that products used in fire protection must meet the relevant legal requirements in a given country and be adequate to the fire situation. In addition, such products must have a certificate of approval issued by the appropriate approval body.

## 2. Powder Extinguishing Agents

The EN 615 standard [8] defines dry powder as an extinguishing medium in the form of finely divided solid chemicals, consisting of one or more main components and additives improving its properties [9]. The main advantage of powders is the combination of physical and chemical action on the fire, which makes the extinguishing process relatively quick. It should be noted, however, that powders indicate a negative impact on the environment, cause losses after a fire due to the elements being covered by a cloud of powder, and pose a threat to humans through the possibility of getting into the eyes or having an allergic effect on the skin. Moreover, they are characterized by a high price.

Depending on the criterion, several divisions can be found. One of them takes into account the division into classes of fires for which they were intended. It includes “BC” and “ABC” powders as well as “D” special powder intended for extinguishing class D fires, i.e., metal fires. Since powder extinguishing agents are compositions containing powdered chemical compounds characterized by flame inhibition, the extinguishing capabilities of powders depend on their chemical composition and determine the use of a given powder for extinguishing certain types of fires (Table 1) [10].

The effectiveness of extinguishing powders is related to their properties, in particular the ability to deactivate free radicals, which then affects the interruption of the chain combustion reaction. Additionally, the low melting point is important. It allows the powder to melt quickly, and form a protective layer on the surface of the burning material, reducing oxygen access to the burning surface. What is more, the access of oxygen is limited by the clouds of extinguishing powder formed above the burning surface. At the same time, with a sufficiently high pressure of the powder stream, the flames are mechanically blown out. It should be noted that alkali metal salts are characterized by a very good ability to inhibit free radical reactions; however, the extinguishing efficiency of salts containing the same anion depends on the presence of the appropriate alkali metal cation and decreases with its atomic mass [11]. Still, potassium and sodium are the most frequently used for economic reasons. The use of a given alkali metal salt determines the extinguishing efficiency, which varies with the anion used, according to the series: PO_4_^3−^ < SO_4_^2−^ < Cl^−^ < Br^−^ < J^−^ < CO_3_^2−^ < CN^−^ < C_2_O_4_^2−^. Salts containing oxalates and cyanides are not used due to their strong toxic properties [11].

Carbonate powders are used to extinguish class B and C fires, while phosphate powders are used to extinguish fires of solids, flammable liquids and gases (ABC). Chlorine powders, on the other hand, based on metal chlorides (most often NaCl or KCl), are used to extinguish metal fires. The greatest efficiency in extinguishing fires is shown by the BC type extinguishing powders, i.e., powders based on carbonates and urea powders. The salts of these powders, such as sodium hydrogen carbonate NaHCO_3_, have an inhibitory effect on the flame, thus slowing down the course of free radical reactions during a fire. ABC type extinguishing powders, based on phosphate salts, also create a glassy layer on the surface of the combusted solid material, isolating the fuel from the oxidant. The separating layer is formed by the thermal decomposition of phosphate salts and the formation of polyphosphates [12]. A feature of ABC powders containing ammonium dihydrogen phosphate (NH_4_)H_2_PO_4_ is hindering the process of reignition of cellulose materials. Metaphosphoric acid (HPO_3_)_3_, which has a low melting point, appears when the dust comes into contact with a flame, forms a glassy film on solid surfaces and cuts off contact with oxygen in class (A) fires that burn with embers [13]. This process is responsible for the fire suppression effect. The dry powder also has a cooling effect. However, the thermal energy required to decompose dry chemical powders is strongly related to the extinguishing capacity of the substance. As a result, all dry chemicals need to be heat sensitive in order for the substance to become chemically active and absorb (“swallow”) the heat. Additionally, spraying of dry chemical powder creates a dust cloud between the flame and the fuel [14]. This cloud, to some extent, isolates the fuel from the heat emitted by the flame.

The inhibitory effect of dry powders occurs according to two processes, i.e., homophasic inhibition or heterophasic inhibition. In the first case of homophase inhibition, the extinguishing powder acting as the inhibitor, as well as the fuel and the oxidant, are in the same gas phase [15]. In the second case of heterophase inhibition, the inhibitor, i.e., the extinguishing powder and its decomposition products, as well as the reactants, i.e., the fuel and the oxidant, are in the gas phase. The inhibitor may be gaseous metal hydroxide or gaseous metal atoms. The gaseous metal hydroxide is produced by the pyrolysis of oxygen salts in a two-step reaction to form a liquid metal oxide which then reacts with water vapor [15,16,17]. On the other hand, anaerobic salts decompose in the presence of hydrogen ions to form metal cations which, when reacted with water vapor, form metal hydroxide. The latter reaction is reversible, which explains the lower inhibitory effect of anaerobic salts in extinguishing flame fires. It is, in fact, a chain reaction that limits the combustion process. This is possible due to the strong reducing properties of alkali ions that arise through the recombination of these ions until they react with carbon dioxide. In this case, the inhibition process takes place with the formation of many transition complexes [15,16].

In the process of heterophasic inhibition, the extinguishing powder is in the solid phase and the reactants (fuel and oxidant) in the gas phase. Radicals present in the gas phase collide with solid powder particles; this in turn prevents their further participation in the combustion process due to energy loss as a result of the collisions. Such a course of the process causes the advantage of recombination reactions over propagation reactions, and the combustion process is inhibited [15,16].

Another mechanism influencing the effective operation of extinguishing powders is the isolation of the fuel from the oxidant. This mechanism is characteristic of fire-extinguishing powders containing salts of phosphoric acid [18,19]. Such powders are particularly useful in extinguishing fires of solids forming glowing coals, classified as group A, which takes place during the combustion of materials of organic origin. Extinguishing powder containing orthophosphates when applied to a burning surface undergoes a cycle of changes leading to the formation of poly metaphosphoric acid. Under fire conditions, it is a liquid substance of high viscosity, which, after the fire stops, forms a glassy layer that insulates the fuel from the oxidant. This layer significantly reduces the risk of glowing coals reignition (e.g., in the case of cellulosic materials).

An important influence on the effectiveness of the extinguishing powder, apart from the composition, which is a mixture of various metal salts and water-repellent agents, is also the size of the powder grains. Smaller powder particles increase the surface area and break down particles of various substances faster with the release of decomposition products neutralizing free radicals. Consequently, they provide better interaction with the burning material and positively influence the fire extinguishing capacity. It was found that the powders with various particle sizes were the most effective, i.e., containing:–from 60 to 80% by weight of the total powder in the range of 20 ÷ 60 μm;–from 10 to 15% by weight of the total powder in the range of 100 ÷ 200 μm [20].

It was also found that the content of the fraction with a particle size of 100 ÷ 200 μm positively influences the range of the powder stream and the effectiveness of its penetration into the fire [20]. On the other hand, a too small particle size of the powder may cause entrainment of these particles by volatile substances released during a fire, which may determine a reduction in extinguishing efficiency [15,16].

The characteristics and structure of the materials used are changing faster and faster. The requirements regarding the need to ensure greater security and better and more effective security are also changing. All this requires work aimed at increasing the effectiveness of extinguishing powders, including the modification of extinguishing powders with nanoparticles immobilized in the structure of extinguishing powder particles. Ni et al. [21] developed a new type of extinguishing powder by immobilizing NaHCO_3_ nanoparticles on porous zeolite [21]. The new powder based on potassium salts with organic and inorganic additives was prepared using a ball mill [22]. The results showed that the modified powders are superior in fire suppression compared to commercial products. Research work focused on the use of magnesium hydroxide, including those containing nanoparticles [23]. It has been shown that the developed powder is characterized by higher extinguishing efficiency compared to commercial BC and ABC powders. Very good results of the extinguishing efficiency were obtained for the extinguishing powder consisting of nanoparticles of magnesium hydroxide with the addition of melamine cyanurate and phosphorus-based ODOPB (phosphorus ODOPB—10-dihydro-9-oxa-10-phosphaphenanthrene-10-oxide) [24]. The components of the extinguishing powder were mixed wet, dry and ultrasonically in order to increase the homogeneity of the nanocomposite. The fire extinguishing time with the nanocomposite was 45.2% shorter than that of the commercial ABC–MAP powder [24]. Moreover, the amount of nanocomposite used was 63.2% lower than in the case of commercial powder. The possible mechanism of action of this nanocomposite fire extinguishing powder was also discussed. On the basis of the results obtained, it was found that the extinguishing mechanism in this case is complex and consists of simultaneous chemical and physical processes as well as a cooling and flame suppression effect [24].

It should be noted that dry chemical powders have good stability at low and normal temperatures. At higher temperatures, some additives melt and cause stickiness; therefore, it is recommended that the storage temperature not exceed 50 °C. At fire temperatures, the active substances contained in the dry chemical powder decompose and perform their extinguishing functions. Apart from basic substances, the composition of extinguishing powders also includes various additives that are responsible for improving the performance of fire extinguishers, including resistance to wetting with water, fluidity, resistance to caking and facilitating encapsulation. Literature reports indicate that work is still underway to improve the extinguishing properties of extinguishing powders.Nie et al. [25] introduced nanometer-sized, very-fine sodium bicarbonate powder into the pore structure of zeolite to solve the problem of powder agglomeration. This allowed the powders to penetrate the flame efficiently and achieve better extinguishing performance [25,26]. Whereas Kuang et al. [22] found that the extinguishing performance of the ultra-fine powder was related not only to the size of the powder surface structure, but also to the main constituents of the extinguishing agent. The experiment showed that the nanometric magnesium hydroxide powder had a higher extinguishing efficiency than the commercial dry ammonium hydrogen carbonate powder [26,27]. Koshiba et al. [28] conducted an experimental study of combustion inhibition and found that metallocene powder has greater fire-fighting benefits than the commonly used ammonium dihydrogen phosphate. The minimum extinguishing concentration of ferrocene was 11 times lower than that of ammonium dihydrogen phosphate, indicating that the metallocene, represented by ferrocene, was more effective. The effect of compounds containing manganese and zinc on the combustion rate of a methane–air flame was investigated experimentally by Linteris et al. [26,29] and compared with iron pentacarbonyl Fe(CO)_5_ and CF_3_Br bromotrifluoromethane. It was found that powders containing nanoparticles of manganese or zinc were more effective in extinguishing the flame, and the effectiveness of stopping fires of zinc compounds was twice as high as CF_3_Br [26].

A typical extinguishing powder available on the market (e.g., ABC extinguishing powder: specific surface 0.34–0.46 m^2^/g) may prove insufficient due to the sedimentation time and small contact surface, e.g., when extinguishing a gas flame (Figure 3). The solution for extinguishing this type of fire are nanopowders (grain diameter approx. 100 nm, specific surface area 25–100 m^2^/g), which fall in the air at a speed of 7.3 cm/day, creating an aerosol cloud, thanks to which the extinguishing efficiency is approx. 30 times greater than common powder [30,31]. However, it should be remembered that not all nanopowders have quenching properties, an example of which is a powder containing NiO nanoparticles. For extinguishing purposes, ZrO_2_ works well, as it inhibits methane combustion. Extinguishing aerosols are also an interesting solution, i.e., fire extinguishing agents produced as a result of the reaction of solid combustion in special aerosol generators. This method does not reduce the oxygen level in the air. The task of extinguishing aerosols is to bind free radicals resulting from combustion processes with a highly efficient and effective extinguishing aerosol. Its micro- or nanoparticles present in active surfaces break the chain of physicochemical reactions. The use of these measures has an impact on the environment, as they leave a trace of contaminants at the site of the fire. In the products generated in the production of aerosols, there are substances such as ammonia, nitrogen oxides, carbon monoxide and hydrogen cyanide which have a harmful effect on the human body and the environment [32].

Although the extinguishing aerosols do not contain any corrosive substances, in an aqueous environment they can reach a pH of 8–10, which can damage materials and equipment sensitive to high pH. Due to the alkaline reaction of the depositing aerosols after discharge, they are not recommended for use in “clean rooms”. It should also be ensured that at high temperatures in the stream, the firefighter does not get burned during the discharge. An additional hazard may be the release of metal nanoparticles and metal oxides during the event. Many of these compounds are highly reactive [33]. 

Fire extinguishing sprays are based on potassium carbonates and nitrates. They have a grain size in the range of 200 nm and at aerosol release temperatures, i.e., > 1000 °C, they can be active centers and react immediately with radicals supporting the flame combustion reaction [32,34].

The aerosols are applicable to fires of class A, B, C and F and electrical devices with voltage up to 36 kV. They are also used during fires in archives and monuments (especially with difficult access), the armaments industry and in the event of maritime incidents. They are produced by extinguishing aerosol generators, which can be part of a permanent fire extinguishing system or constitute a stand-alone extinguishing device, and therefore must meet certain requirements [35]. Depending on how the fire generators are triggered, the aerosol may act locally or volumetrically. Its effectiveness may also vary depending on environmental conditions. The developed surface of the solid and the type of chemical reaction are also important. In the case of nanoparticles, which have a significantly increased surface area, a higher fire suppression efficiency is observed than in the case of particles of relatively larger sizes. However, the reduction of the particle size affects the possibility of agglomeration of the particles, which may result in the difficult contact of the particle surface with the flame radicals. Biel et al. [32] also found that one of the parameters determining the atomization of the nanosize of an aerosol is environmental humidity, which has an effect, inter alia, on the grain distribution and thus on the surface activity of the FP-40C type extinguishing aerosol [32]. The results of the research showed that in dry (30% humidity) and very humid (90% humidity) environments the number of aerosol nanoparticles decreases with time due to both aggregation and lumping, and the most favorable conditions for extinguishing aerosol atomization occur in an environment with a humidity of approx. 70%. This may be related to the more effective capture of water radicals from chain reactions taking place in the flames [32].

Common additives to extinguishing compositions are also silicas [36] (flame or precipitated), metal stearates, talc [34]. The search for the optimal content of functional additives in powder compositions is quite a difficult task. Too low a proportion of the additive causes deterioration of the utility properties of the powder and shortens the guaranteed shelf life. If the amount of the functional additive is excessive in relation to the surface area of the powder particles, the flow of the composition deteriorates due to the increase in the number of contacts and the additional frictional forces acting between the additive particles [37,38]. The data obtained under dynamic conditions make it possible to predict the behavior of powders at high flow rates [39]. The effectiveness of hydrophobized nanosilica with a particle size of about 65 nm as an additive to ammonium phosphate fire extinguishing powders was a subject of a study by Shamsutdinov et al. [40]. The surface of the extinguishing component was evenly covered with spherical hydrophobized nanosilica nanoparticles. The addition of nanosilica made it possible to obtain a hydrophobic coating on the particles of the extinguishing powder. The apparent contact angle of the coated particles was found to be greater than 160°. Dynamic flow resistance of commercial and tested extinguishing powders was compared. The specific energy and flow energy of the aerated test powder were relatively low, which indicates a poor aerodynamic interaction between particles in dynamic processes. It was found that the spray resistance of the tested powder was the lowest among the tested extinguishing powders (including commercial ones), and therefore it was characterized as having the best flow. Similar results were obtained with the use of mesoporous nanosilicas of various structures (MCM-41, MCM-48 and SBA-15) with a large specific surface of over 1400 m^2^/g [41]. Such nanosilicas have been found to improve the fluidity of ammonium phosphate-based extinguishing powders. 

The aforementioned nanosilica positively influenced the improvement in fluidity of extinguishing powders based on ammonium phosphate. It has been shown that with the reduction of the particle size of the powder (or agglomerates), the internal friction between the powder particles decreases [42,43].

The particle size distribution is also important. It has been found that the narrow particle size distribution makes it possible to obtain a powder with better flow properties [44]. Veregin and Bartha [45] showed that the particle size of the functional nanoadditive affects the flow properties of powder materials and depends on the forces of interparticle interactions and the contact radius of the powders’ surface.

The reduction in particle size is accompanied by a reduction in their contact surface, which leads to a decrease in the influence of interparticle forces. It was found that increasing the mixing time of the powders improves the properties of the powders [46]. This effect results from the possibility of obtaining an increase in the homogeneity of the coating of extinguishing powder particles with silica nanoparticles and reducing the possibility of agglomeration of silica nanoparticles.

It has been found that the use of hydrophobic additives significantly improves the fluidity of the materials by eliminating capillary bridges between the particles, thus ensuring easy breaking of agglomerates of particles in powder compositions [47,48]. It should be emphasized that if the amount of silica nanoparticles exceeds the surface area of the powdered extinguishing material, the flow properties of the composition decrease due to the increasing contact and friction forces between the nanoparticle and particles [49,50].

Modification of the properties of extinguishing powders by introducing nanoparticles enables a significant increase in their extinguishing efficiency (Table 2) and makes it possible to improve the safety of people and property during an incident. 

The introduction of nanoparticles to the extinguishing powder changes the properties of the extinguishing powder by:
–changing the composition of the extinguishing powder particles after incorporation of nanoparticles, which modifies the powder’s mechanism of action and enables the extinguishing efficiency of such powder to be increased;–immobilization of nanoparticles on the surface of extinguishing powder particles, which changes their physical properties, such as fluidity or hydrophobicity, significantly improving the extinguishing efficiency of the powder.

It should be noted that nanosubstances are released into the environment, characterized by diverse activity, which is conditioned by the size of the surface, and the shape and size of the grain or structure. Therefore, they can have various impacts on human health and environmental safety. It is therefore necessary, when introducing new solutions, to carry out an assessment in the field of toxicology and biodegradation or interaction with compounds present in the environment.

## 3. Foam Concentrates

Foam concentrates used in rescue and firefighting operations are used to generate foam, i.e., bubbles made of liquid. The EN 1568 series of standards specifies the requirements for the physicochemical properties and minimum effectiveness of extinguishing foams intended for the production of medium-, light- and heavy-expansion foams suitable for surface application to liquids immiscible and/or miscible with water. The EN 1568 standard [51] distinguishes types of foam extinguishing agents such as: synthetic (S), protein (P), fluoroprotein (FP), alcohol-resistant (AR), aqueous film-forming concentrates (AFFF), fluoroprotein water film-forming agents (FFFP) and fluorine free foam concentrates (F3).

The composition of synthetic foaming agents (S) is based on surfactants that do not contain organofluorine compounds. Protein agents (P) contain hydrolysed protein substances of animal origin, and fluoroprotein (FP) and film-forming fluoroprotein concentrates (FFFP) additionally fluorinated surfactants. The composition AFFF is based on mixtures of hydrocarbons and fluorinated foaming agents, thanks to which they have the ability to form a water film on the surfaces of some fuels. Alcohol-resistant foaming agents (AR) in their composition contain a mixture of the above agents, additionally containing substances that make the foams resistant to polar liquids having the ability to form a polymer film on the alcohol surface. Fluorine free foam concentrates (F3) are intended for applications analogous to AFFF and/or AR foams. They are based on mixtures of hydrocarbon surfactants and non-fluorine stabilizers [52]. The mixtures also contain corrosion inhibitors and substances that lower the freezing point [53,54].

Pursuant to the Regulation of the European Union Commission [55], from July 4th 2020, the following may not be manufactured or placed on the market: foaming agents containing perfluorooctanoic acid (PFOA) or its salts in a concentration equal to or higher than 25 ppb PFOA, including its salts or 1000 ppb of one derivative or a combination thereof. The environmental problem of using hazardous substances is solved by adding other surfactants that do not contain fluorine or by adding a small amount of short chain fluorocarbon surfactant to replace the long chain fluorocarbon surfactant. 

Foam is used both to extinguish fires, mainly class B, but also class A. In the event of a fire of polar liquids, it is necessary to use foam produced by foam-producing agents specially developed for this purpose. Fire extinguishing foams are completely useless for extinguishing gas fires. Additional hazards may arise when extinguishing fires of substances that react with water, such as organoaluminum compounds, metals or carbide [56]. Therefore, fire foams are primarily used to extinguish flammable liquids in chemical, food or coke industry plants, as well as refineries and petrochemical plants.

The properties of extinguishing foams are determined by the ratio of the foam volume to the volume of the solution from which the foam was formed, expressed by the so-called foaming number. The use of appropriate foaming agents makes it possible to obtain light-, medium- and heavy-expansion foam (Figure 4).

The properties of foams are also determined, in addition to the number of foaming, by parameters such as dispersion, foam durability or fluidity. The degree of disintegration of the foam bubbles determines the dispersibility. The test results showed that the larger the bubble diameter, the smaller the dispersibility [57]. The ability of the foam to maintain its properties, which it obtained at the time of production, is determined on the basis of the draining time of the aqueous solution of the foaming agent therefrom. Durability depends not only on the number of foaming or dispersibility, but also on such elements as the concentration and properties of the foaming agent, and the quality of the water which is the basis for the production of the foam. The study results showed that the slower the bubble destruction process takes place, the more durable the foam, and the faster it spreads over the surface of the burning material, the less foam is destroyed and the faster fire is extinguished [57,58]. Therefore, when developing the agent, one should also take into account shaping the behavior of the extinguishing foam depending on the temperature of the fire and air, and the type of environment with which the foam comes into contact [57,58].

Fire foams, thanks to the creation of a low surface tension of the water solution and the foam concentrate, allow the water film, although heavier than the flaming liquid, to freely spread over its surface. This prevents flammable gases and vapors from entering the combustion zone, allowing the fire zone to cool down. A properly applied layer of foam allows for the elimination of the combustion zone and the cooling of the fire zone. Therefore, the effectiveness of a fire extinguishing action with the use of foam extinguishing agent depends not only on its proper application, but also on the quality of the foam concentrate [58].

Therefore, it should be stated that the extinguishing foam is a thermodynamically unstable dispersion system and the foam stability in difficult conditions is still an important element determining the effectiveness and safety of its use. Therefore, work is underway on the stability of foam extinguishing agents by introducing new agents, including nanoparticles [59]. For example, Li et al. [60] used a titanium dioxide (H-TiO_2_)/gel system with a three-dimensional lattice structure and a flame retardant effect. H-TiO_2_ foam gel was obtained by mixing acrylic acid (AA) and acrylamide (AM) with sodium borate and H-TiO_2_, the optimal gelation time was 18 min at 40 °C, and the concentrations of the cross-linking agent and initiator were 0.3 wt.% and 0.4 wt.%. The results obtained show that for the coal sample the ignition time (TTI) of the H-TiO_2_ foam gel was up to 12 times longer than for the raw coal samples, while the maximum heat release rate (HHR) was reduced by 56.59%, and the minimum total emission heat (THR) was reduced by 32.04% [60,61].

Aluminum hydroxide nanoparticles (Nano-ATH) were used as an additive to the composite foaming agent solution to prepare nano-ATH foam with high stability [62]. The results of the work indicated that after the addition of nano-ATH, the viscoelastic modulus of the foam liquid film increased from 0.91 mN/m to 3.08 mN/m, and the foam volume increased from 380 mL to 700 mL. There was also an increase in the mechanical strength of the liquid film at a mass concentration of 2%, as the half-life of the foam increased from 78 sec to 453 s. In the samples of carbon with nano-ATH foam, the ignition time increased from 26 sec to 176 sec, the combustion sustained time was reduced from 520 sec to 211 sec, the peak heat release rate decreased from 117 to 58 kW/m^2^ and the peak smoke production rate decreased from 0.084 to 0.0049 m^2^/s during combustion, indicating good flame retardant properties and smoke suppression [62].

Additionally, nano-foam extinguishing agents with a non-toxic LC50/LD50 certificate have been developed. They are characterized by high resistance to oil and fire, as well as effective insulation when mixed with water to extinguish an electric fire [63]. The size of the nanoparticles used in the devices ranges from 80 to 150 nm on average. Due to the addition of a fluorosurfactant, the foam has a good fluidity and the nanofoam extinguishing agents are sprayed with high pressure gas to form a thin film on the surface of the oil in case of oil contamination. Thanks to the material refinement technology used, the foam gets better parameters and covers a larger area, and thus effectively blocks gas combustion, suppresses fuel vapors after extinguishing a fire and prevents the backflow of the air.

The idea of using nanoparticles was used in the research on the improvement of aqueous film-forming concentrates (AFFF). A structure has been developed that consists of 3–12 parts by weight of a complex surfactant, 10–150 parts of nanoparticles, 0.3–1 parts of stabilizer, 15–30 parts of dispersant, 3–9 parts of antifreeze, 1–10 parts suspending agent and water [64]. For the preparation of the suspension, nanoparticles of silicon dioxide, aluminum hydroxide, aluminum oxide, magnesium hydroxide, magnesium oxide, antimony oxide, calcium carbonate, iron oxide and antimony trioxide were used. To prepare the foam, the nanoparticles in the form of a suspension were evenly mixed with the complex surfactant solution. The extinguishing agent was added with a short chain fluorocarbon surfactant or a complex surfactant combined with an organosilicon surfactant and a hydrocarbon surfactant to form a compact foam layer and a closed water film layer, and obtain adequate stability and thermal insulation of the foam. The addition of nanoparticles has improved the thermal insulation properties of the foam layer and has avoided rapid cracking of the foam caused by too high an external temperature. Appropriate dispersion and stability of the foam containing nanoparticles dispersed in other components has a positive effect on the extinguishing efficiency and control of solid and liquid fires [64].

It should be added that silica is widely used in foams, which is characterized by a large surface and the ability to absorb water. The foam covers the surfaces, protecting them against the spread of the flame front, and separating the fuel from the oxygen. Additionally, low-expansion foams provide significant cooling [65,66,67]. Large-size ultrafast gelled foams have been developed, which are characterized by exceptional thermal stability, mechanical strength and biocompatibility. They allow the production of controlled gelling hybrid silica foams in the viscosity range from 2 to 30 Pa·s to 100 Pa·s. The foam consists of ordered silica nanoparticles with a narrow particle size distribution of ∼10–20 nm. The obtained results indicate that the extinguishing efficiency of silica-based sol-gel foams is almost 50 times higher than that of ordinary water. Importantly, the biodegradation index is only 3 days, while even for conventional foaming agents this index is several times higher [68].

Strong foam stabilizing effect with the use of SiO_2_ nanoparticle-cationic surfactant tetradecyltrimethylammonium bromide (TTAB) mixtures was obtained at concentrations of SiO_2_ nanoparticles as low as 2 wt% [69]. Binks et al. [70] found that silica nanoparticles, modified with chloro- and methyl- silanes, can stabilize the foam even for several days without disproportionation. Solid particles increase the flow resistance of the fluid in the liquid film, so they can stabilize the foam [71,72].

Work is underway on foams used in the event of fires of hydrocarbon fuels. Current fire protection formulations, based on perfluorinated surfactants, are highly toxic and have extremely poor biodegradability, spread through the food chain and affect many species over an extended period of time. The work covers composites of metal carbonate nanoparticles and liquid ionic surfactants, surfactants in the form of nanocapsulated metal carbonate ions (NEIL), so that it is possible to create a stable foam and release CO_2_, a compound depriving fire of oxygen [73,74].

The analysis of the results of work on the application of nanotechnology for the modernization of foaming agents showed that silica nanoparticles are most often used in the works (Table 3).

The addition of nanoparticles improves the properties of fire-extinguishing foams in terms of stability, extinguishing efficiency, and smoke suppression. and thus improves the safety of both people, the environment and property. However, fire extinguishing foams may, to a greater or lesser extent, affect the environment and human health. The presence of synthetic surfactants and other chemical substances constituting the foam composition may have an irritating and toxic effect on humans and constitute a significant environmental pollution. Nanoparticles can also have a significant impact on humans and environmental processes. As indicated with surfactants, non-polar substances, such as hydrocarbons, which are persistent carcinogenic compounds, may migrate to soil and groundwater, posing a threat to human health and the quality of the environment. Therefore, the modifications should consider not only the effectiveness in terms of effective and quick fire extinguishing, but also minimizing the risk to environmental safety. 

## 4. Other Fire Extinguishing Agents

The most frequently used extinguishing agent, mainly due to its availability, low price and relatively good extinguishing properties, is water. The extinguishing property of water is related to its cooling effect. It lowers the temperature of the burning material and the combustion zone. The water vapor generated during the extinguishing inhibits the reactions of free radicals with flammable gases, thanks to which it dilutes the combustion zone and has an insulating effect [75].

A contraindication to the use of water to extinguish fires are those fires of flammable materials which react violently in high temperatures in contact with water, causing additional danger. These are, i.a., fires of metals which react with water at room temperature or at combustion temperature, releasing large amounts of heat [15,76]. The increase in the extinguishing efficiency of water is possible by addition of chemical compounds that change its physical properties. The purpose of these compounds is to lower the surface tension of water, which results in better absorption of water into the surface of solid combustible materials. The compounds lowering the surface tension of water include, among others, wetting agents, i.e., compounds containing in their molecule a hydrophilic chain saturated with fluorine atoms, attached, for example, to the potassium salt of an aliphatic acid sulphamide, which are added to water in an amount of 0.5 to 2%, thickeners (15 to 25%) or thickeners—wetting [77].

For example, Dali et al. [78] showed that the quenching time of liquid hydrocarbons by suspensions containing carbon nanostructures (CNS), such as functionalized multi-walled carbon nanotubes (MWCNTs), is on average 3.5–5.0 times shorter than the quenching time of liquids with finely divided water. Wetting agents and additives increase the intensity of the heat sink, creating a film on the surface of the burning oil product. Carbon nanotubes increase the thermal conductivity and change the rheological properties of liquids at low concentrations [79]. It has also been found that the CNS water-based slurries are extinguishing agents with a predominantly cooling and diluting effect. Suspension droplets enter the combustion zone, which causes an intense heating to the boiling point. Such a process causes evaporation and cooling of the combustion zone, and the flame is extinguished with a sufficient amount of water vapor in the combustion zone [78,80]. However, it should be remembered that too high a concentration of nanoparticles may lead to aggregation of nanoparticles, which reduces the effective thermal conductivity of the suspensions and the specific heat of vaporization.

It should be noted that due to the good fluidity of the water, most of it is lost when sprayed onto the fire site. Large amounts of water are required for large-scale fires, rapid fire spread, and when extinguishing poses significant difficulties. Often, in the case of strong winds and the need to drop the agent from a height, it is hard to control the effectiveness of operations and some water resources are wasted. Therefore, research is being carried out on water retention in appropriate structures. Such a solution is, among others, superabsorbent resin, which is a three-dimensional network polymer. It has hydrophilic groups in its structure and is slightly cross-linked, which means that it can absorb a large amount of water, swell and keep water from flowing out. High-molecular polyacrylic acid superabsorbent resin hydrogel is widely used in the field of firefighting. This substance has a significant heat capacity under high-temperature conditions. However, some superabsorbent resins have a low self-adhesive capacity, and have difficulty in adhering to vertical wall surfaces, wooden furniture surfaces or surfaces containing stainless steel such as oil tanker surfaces. These objects cannot, therefore, be effectively covered, which prevents successful extinguishing and fire protection. To improve these properties, a polymeric hydrogel extinguishing agent has been developed. It contains 0.1–0.5 wt. % of a polyacrylate superabsorbent resin and/or an acrylate–acrylamide copolymer, and 0.01–0.5 wt. % inorganic nanoparticles and water. Inorganic nanoparticles, mainly diatomite, kaolin, bentonite, attapulgite, colloidal silica and precipitated silica, added to a special superabsorbent resin improve its viscosity, adhesion of hydrogel extinguishing agent and extinguishing effect [81].

Water can also be used in the form of water mist. Pressurized water is transported through pipes to special nozzles, which then spray water in the form of mist over the area. The water mist extinguishing system allows the temperature to be reduced quickly, the humidity to increase and prevents the oxygen fire from finding a new source. The developed Water Mist technology allows the production of water nanoparticles that facilitate the fogging process and reduce the amount of water needed to extinguish the fire [82]. Very-fine water mist can suppress the flame in case of fires covered with obstructions in tunnels, thanks to better fluidity and obstacle avoidance [83,84]. Kudo et al. [85] investigated the effect of frequency on the size distribution of ultrasonic fog and proposed a mechanism for generating nanometric ultrasonic fog based on the amount of water vapor around the liquid column. It has been found that there is a relationship between the fog size distribution and the ultrasound frequency. Increasing the power intensity and density by changing the surface diameter of the ultrasonic oscillator influenced the numerical concentration and size distribution of the nano-sized fog. Using this technique, the diameter of the fog can be controlled by changing the frequency of the ultrasonic transducer [85].

Similar results were obtained by Jiang et al. [84], who conducted experimental and numerical studies of the effect of nanoparticle water mist with phosphorus-containing compounds (PCC) on the flame in the event of a CH_4_/coal dust explosion. The additives used were dimethyl methylphosphonate (DMMP) and phytic acid (PA). The results obtained showed that the water mist in the form of nanoparticles can act as a thermal barrier and effectively reduce the flame temperature. A system containing DMMP, compared to PA, can effectively lower the concentration of toxic (CO) and flammable gases (CH_4_ and H_2_) in CH_4_ hybrid and coal dust explosions. Additionally, water mist containing DMMP can significantly slow down the speed of the flame and cut off the flame. The results of the research also showed that PCC additives can also increase the heat absorption capacity of the water mist [84].

In order to improve the damping capacity of water mist in the form of nanoparticles, various chemical additives in different concentrations are added to the water. The conducted research allowed for the selection of various inorganic salts and organic compounds, including compounds containing Na, K, P, Ca, Fe and surfactants. According to experiments and computational studies, it has been proved that the addition of salt additives can improve the effectiveness of chemical suppression [84,86,87,88].

Another solution in the field of firefighting is a fire blanket, handheld firefighting equipment used to mechanically cut off the air supply to burning materials. Its use involves tightly covering a small, burning object, e.g., a container with a burning substance or burning clothes. When using a blanket, however, one should remember to cover the source of the fire from his or her side to avoid burning. At the same time, it should be noted that the blanket can be used effectively only to extinguish small sources of fire located close to the person putting out the fire. However, it is possible to use it multiple times and not to destroy the extinguished objects. High-temperature fire blankets are made of non-flammable or fire-resistant fabric materials (e.g., aramids, glass fiber, amorphous silica, pre-oxidized carbon and mineral fibers) [89]. Fire blankets made of polyester or wool are useless in the event of a fat fire, as they are at risk of spontaneous combustion or ignition. Blankets can also be made of fiberglass in the form of nano fiberglass. The Nano Fiberglass fabric [90] helps to extinguish the most difficult flames, regardless of the cause, in a very short time. It can be used to extinguish all types of fires, including fires of lubricants, liquids and gases.

An interesting solution is the invention [91], which allows the use of nanocrystalline particles with a relatively large surface area to reduce the amount of various substances generated during fires and to suppress the fire itself. The results of the research showed that nanocrystalline particles can come from the group consisting of metal oxides, metal hydroxides, carbonates, bicarbonates, phosphorus, inorganic phosphorus compounds, boron compounds, antimony compounds, molybdenum compounds, titanium compounds, zirconium compounds, zinc compounds, amidosulfonates, sulfates, bromine compounds, chlorine compounds and mixtures thereof. Metal oxides and metal hydroxides Mg, Sr, Ba, Ca, Ti, Zr, Fe, V, Mn, Ni, Cu, Al, Si, Zn, Ag, Mo, Sb and mixtures thereof are the most preferred nanocrystalline materials. However, sodium, aluminum, magnesium and calcium hydroxides, carbonates and bicarbonates are most preferred. It has been shown that the size of the nanocrystalline particles should be less than 25 nm; nonetheless, the most optimal particles are about 1–20 nm, especially between about 2 and 10 nm. In contrast, the values characterizing the multipoint Brunauer–Emmett–Teller (BET) surface area should be at least about 15 m^2^/g, with the most optimal being at least about 70 m^2^/g and most preferably in the range of about 200–850 m^2^/g [91].

The effectiveness of smoke removal is determined by the amount of nanocrystalline particles dispensed in a given area (i.e., mass concentration of nanoparticles), aerodynamic geometric mean diameter (GMD) of the particles and the settling velocity of the particles. The most optimal amount of nanocrystalline particles to be dispensed in a region is between about 1–5 g/m^3^. The nanocrystalline particles at first must be scattered in the area affected by the smoke to absorb some of the smoke, especially the carbon particles, which tend to obscure visibility. The developed solution also allows a reduction of the amount of various toxic compounds, such as acrolein, toluene diisocyanate, formaldehyde, isocyanates, HCN, CO, NO, HF and HCl generated during a fire [91].

Nanotechnology creates opportunities to improve the effectiveness of firefighting (Table 4) and thus the safety of users and people involved in the event.

The use of nanoparticles shortens the fire extinguishing time, reduces the concentration of toxic and combustible gases and reduces the amount of dust, which significantly reduces pollution and migration of pollutants along with smoke over longer distances. It should also be noted that the fragmentation of water particles to sizes < 100 nm allows a reduction of the amount of water needed to extinguish the fire and the amount of fire extinguishing sewage.

## 5. Conclusions

The fire protection system must be constantly improved as a consequence of changes taking place in the economic, social and legal space. In addition to preventive measures, research is necessary in the field of tools used in rescue operations. Solutions based on nanotechnology are being introduced more and more often. They are aimed at improving the key properties of individual extinguishing agents and thus their effectiveness in a firefighting situation. In this context, the requirements for efficiency improvement play a major role, which are becoming more and more demanding.

The analysis of the type of nanoparticles used in the processes of modification of extinguishing agents shows that inorganic nanoparticles, mainly silica and oxides, hydroxides and bicarbonates/carbonates of alkali metals and aluminum, are the most widely used. The introduction of these nanoparticles to standardly used powders or foaming agents allows for faster fire extinguishing, greater efficiency, reduction in the emission of toxins into the environment and reduction in heat emission to the environment. An interesting solution is also the reduction of water particles to the size of nanoparticles, using, among others, ultrasound, which creates a thermal barrier and reduces the time of extinguishing the fire, as well as reduces the consumption of the extinguishing agent. Thus, it improves the safety of people involved in the event and allows you to reduce the costs of occurring fires.

However, selection of nanoparticles must be appropriate, because at higher concentrations they may be subject to aggregation, which in turn reduces the effectiveness of the quenching process. At the same time, it should be noted that foams and extinguishing powders may have a negative impact on the environment due to the use of various types of chemical compounds with greater or lesser chemical activity in their production. As a result of the application of such measures, some areas can become contaminated and require further treatment, consisting in the neutralization of substances created during the fire and extinguishing, at least. Therefore, in order to further improve the extinguishing efficiency and the resistance of the extinguishing agent, as well as to minimize environmental pollution, it is necessary to develop extinguishing agents that can extinguish various complex fires and, at the same time, remain stable and environmentally friendly.

Solutions implemented on the basis of nanotechnology contribute to faster cooling of the environment, improved visibility in case of high smoke and more effective fire extinguishing action. However, work carried out in this scope and the modifications introduced need to consider the toxicity of nanoparticles, the possibility of their migration and the negative impact both on the environment and on humans, including those involved in the incident.

## Figures and Tables

**Figure 1 materials-15-08876-f001:**
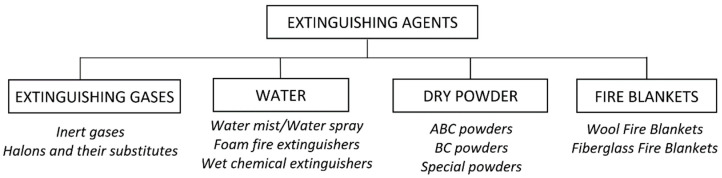
Division of extinguishing agents.

**Figure 2 materials-15-08876-f002:**
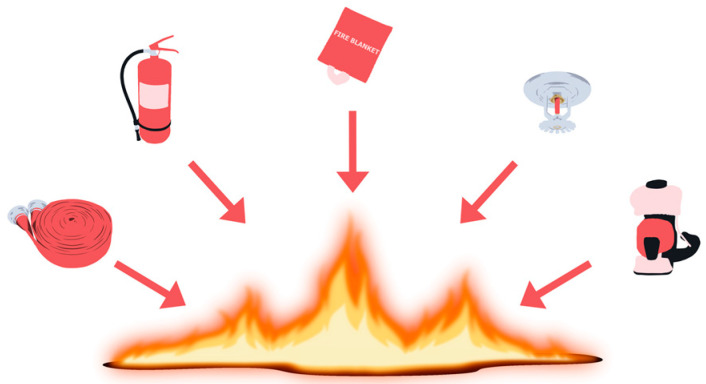
Extinguishing media covered by the work on modification based on nanotechnology.

**Figure 3 materials-15-08876-f003:**
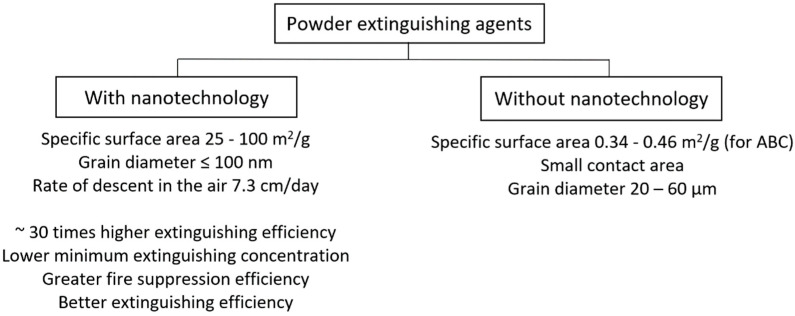
Comparison of standard and modified extinguishing powders with the use of nanotechnology.

**Figure 4 materials-15-08876-f004:**
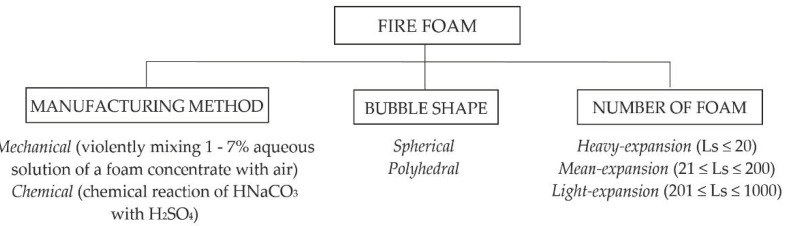
Division of fire foams.

**Table 1 materials-15-08876-t001:** Chemical composition of the base of the powder fire extinguishing agent vs. application to extinguish different groups of fires.

Extinguishing Powder Base	Chemical Formula	Class of Fires Being Extinguished
Potassium hydrogen carbonate	KHCO_3_	BC
Urea + Potassium hydrogen carbonate	NH_2_CONH_2_ + KHCO_3_	BC
Sodium hydrogen carbonate	NaHCO_3_	BC
Sodium tetraborate	Na_2_B_4_O_7_ · 10H_2_O	D
Potassium chloride	KCl	D
Sodium chloride	NaCl	D

**Table 2 materials-15-08876-t002:** Possibilities of using nanotechnology to modify powder extinguishing agents.

Nanoparticles	Characteristic	Effect	Ref.
NaHCO_3_	NP-NaHCO_3_ on porous zeolite	improving fire suppression efficiency	[21,25]
Ferrocen	sized nanoparticles < 100 nm	greater efficiency of extinguishing flames and stopping fires	[28]
Mn, Zn	sized nanoparticles < 100 nm	greater efficiency of extinguishing flames and stopping fires	[29]
Mg(OH)_2_	NP-Mg(OH)_2_ with the addition of melamine cyanurate and phosphorus ODOPB—10-dihydro-9-oxa-10-phosphaphenanthrene-10-oxide	higher extinguishing efficiency, shorter fire extinguishing time by about 45%, lower consumption of extinguishing powder	[24]
SiO_2_	hydrophobized nanoparticles with a size of about 65 nm as an additive to extinguishing powders based on ammonium phosphates	improving the fluidity of powders	[40]
SiO_2_	mesoporous nanoparticles of various structures (MCM-41, MCM-48 and SBA-15), characterized by a large specific surface (>1400 m^2^/g)	improving the fluidity of powders	[41]

**Table 3 materials-15-08876-t003:** Possibilities of using metal nanoparticles and their compounds in extinguishing foams.

Nanoparticles	Characteristic	Effect	Ref.
TiO_2_	H-TiO_2_/gel system with a three-dimensional network structure	flame retardant effect, longer ignition time	[60]
Al(OH)_3_	addition to the composite foaming agent solution, mass concentration 2%	high foam stability, good flame retardant and smoke suppression properties	[62]
SiO_2_, Al(OH)_3_, Al_2_O_3_, Mg(OH)_2_, MgO, FeO, Sb_2_O_3_	evenly mixed with the complex surfactant solution	improving the thermal insulation properties of the foam layer, affects the efficiency of extinguishing and controlling solid and liquid fires	[64]
SiO_2_	ordered nanoparticles with a narrow particle size distribution of ∼10−20 nm	50-times higher extinguishing efficiency, biodegradation index—3 days	[68]
SiO_2_	SiO2-TTAB mixture, with an NP-SiO2 concentration of 2 wt%.	strong foam stabilizing effect	[69]
SiO_2_	NP-SiO_2_ modified with chloro- and methyl- silanes	stabilization of the foam even for several days without disproportionation	[70]
Me_x_(CO_3_)_y_	surfactants in the form of nanoencapsulated metal carbonate ions (NEIL)	formation of a stable foam and the release of CO_2_	[73,74]

**Table 4 materials-15-08876-t004:** Possibilities of using nanotechnology to improve the efficiency of firefighting.

Nanoparticles	Characteristic	Effect	Ref.
CNS	suspension of carbon nanostructures (CNS) in the form of functionalized multi-walled carbon nanotubes (MWCNT)	shorter fire extinguishing time, mainly cooling and diluting effect	[78]
diatomite, kaolin, bentonite, attapulgite, SiO_2_	inorganic nanoparticles added to a special superabsorbent resin	improving the adhesion of hydrogel extinguishing agent, better extinguishing effect	[81]
mist of H_2_O	Water Mist technology	reducing the amount of water needed to extinguish the fire, the ability to avoid obstacles, quickly lowering the temperature	[82,83]
	ultrasonic fragmentation of H_2_O particles		[85]
	system containing dimethyl methylphosphonate (DMMP)	reducing the concentration of toxic and combustible gases and coal dust, the thermal barrier effectively reduces the temperature of the flame	[84]
water glass	nano fiberglass	extinguishing all types of fires (including fires of lubricants, liquids and gases)	[90]
hydroxides, carbonates and bicarbonates of Na, Al, Mg and Ca	nanocrystalline particles with a size of about 2-10 nm, with a surface area of 200–850 m^2^/g	reduction in part of the smoke (especially carbon particles) and the amount of various toxic compounds formed during a fire	[91]

## Data Availability

Not applicable.

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
