# Peer review of "Application of Nanotechnology in Extinguishing Agents"

_materials, 2022, doi:10.3390/ma15248876_

Round 1

Reviewer 1 Report

The authors in this paper reviewed the applications of nanotechnology in extinguishing agents such as powder, foam and others. I don’t see anything significant and suggestive for the reader in the field. I recommend the author to submit the paper to a much more specific journal.

Detailed comments:

(1) The English should be improved.

(2) The following literature is for your reference.

1.       Shinoda K, Nomura T. Miscibility of fluorocarbon and hydrocarbon surfactants in micelles and liquid mixtures. Basic studies of oil repellent and fire extinguishing agents[J]. The Journal of Physical Chemistry, 1980, 84(4): 365-369.

2.       Loboichenko V, Leonova N, Strelets V, et al. Comparative analysis of the influence of various dry powder fire extinguishing compositions on the aquatic environment[J]. Water and Energy International, 2019, 62(7): 63-68.

3.       Rakowska J, Prochaska K, Twardochleb B, et al. Selection of surfactants as main components of ecological wetting agent for effective extinguishing of forest and peat-bog fires[J]. Chemical Papers, 2014, 68(6): 823-833.

Author Response

We would thank the reviewer for taking the time to read the manuscript and commenting. Submission of the article to the journal is due to special issues that allow the development of manuscripts on selected, detailed topics. We have introduced numerous additions and corrections. We hope that the article in its current version will meet the reviewer's expectations in this regard.

Detailed comments:

Point 1: The English should be improved.

Native Speaker verified the prepared text. All changes are marked in red.

Point 2: The following literature is for your reference.

  1. Shinoda K, Nomura T. Miscibility of fluorocarbon and hydrocarbon surfactants in micelles and liquid mixtures. Basic studies of oil repellent and fire extinguishing agents[J]. The Journal of Physical Chemistry, 1980, 84(4): 365-369.
  2. Loboichenko V, Leonova N, Strelets V, et al. Comparative analysis of the influence of various dry powder fire extinguishing compositions on the aquatic environment[J]. Water and Energy International, 2019, 62(7): 63-68.
  3. Rakowska J, Prochaska K, Twardochleb B, et al. Selection of surfactants as main components of ecological wetting agent for effective extinguishing of forest and peat-bog fires[J]. Chemical Papers, 2014, 68(6): 823-833.

We would like to thank the reviewer for pointing out additional references. We read the articles and would like to point out that Literature 1 does not take into account the latest knowledge in this field, therefore it was not included in our considerations. In addition, the indicated literature does not refer to nanotechnology and its application for the production and improvement of the properties of extinguishing agents. However, additional information has been included in the manuscript (in red) based on additional literature data in the field of nanotechnology.

We hope that all done corrections and additions improved the quality of the manuscript.

Reviewer 2 Report

The authors fully reviewed the applications of nanotechnology in extinguishing agents. This review presents the powder extinguishing agents and other extinguishing agents. The English writing of this review manuscript is very good. However, the authors did not use some figures and tables to conclude and describe the application and existed extinguishing agents. I suggest that the authors should added some figures and tables to conclude and present the development of extinguishing agents because it can make the contents to become clear. In general, this review can be published before adding some figures and tables.

Author Response

We would like to thank the reviewer for reading the manuscript and giving positive feedback on the manuscript. Following the reviewer's recommendation, we have introduced additional tables and figures. Native Speaker verified the prepared text. All changes are marked in color.

We hope that all done corrections and additions improved the quality of the manuscript.

Reviewer 3 Report

Please find the attached Review file.

Author Response

We would like to thank the Reviewer for careful and thorough reading of this manuscript and for the thoughtful comments and constructive suggestions, which help to improve the quality of this manuscript. Our responses are as follows:

Detailed comments:

Point 1: The novelty and motivation of the research presented in the manuscript should be emphasised in the last paragraph of the Introduction section.

Thank you for this attention. We introduced the relevant sentence into the last paragraph of the Introduction (lines: 90-96 and 102-106).

Point 2: I suggest summarising the characteristics and findings included in the manuscript in a table. This undertaking will improve readability.

Thank you for this attention. We've inserted the tables into the manuscript.

Point 3:  Move figures and tables into the main body of the manuscript.

As suggested by the reviewer, all graphs and tables have been inserted into the appropriate places in the manuscript.

Point 4: Try to add quantitative results in the abstract to catch the prospective readers' attention. I suggest accepting the paper if the authors adequately address these comments.

As suggested by the reviewer, we added additional information to the abstract (line 15-17).

Point 5: I suggest accepting the paper if the authors adequately address these comments.

We hope that all done corrections and additions improved the quality of the manuscript.

Reviewer 4 Report

The review "Application of nanotechnology in extinguishing agents" is a valuable study of the literature on a very important subject that not always gets the proper attention from the scientific community. It can be published after authors address the following problems:

Abstracts should be more informative, highlight the review concept and scope. As often abstract section is presented separately in search engines, it must be able to stand alone as an informative piece.

While the objective of this review is clear, authors should indicate also the review methodology. The consulted databases, keywords used to retrieve relevant literature studies, time spawn for the query.

The English language needs some minor polishing for style and typos (e.g. “specific to each specific” row 47)

At row 229 correct iron (V) Fe(CO)5 to just iron pentacarbonyl Fe(CO)5. The iron is in zero oxidation state.

Abbreviations must be explained at first use (TTAB row 462, explanation Tetradecyltrimethylammonium bromide I suppose?)

The reading of this review left me with the feeling that on some parts of the manuscript the authors should have presented more information and go in-deep with discussion.

Author Response

We would like to thank the Reviewer for careful and thorough reading of this manuscript and for the thoughtful comments and constructive suggestions, which help to improve the quality of this manuscript. Our responses are as follows:

Point 1: Abstracts should be more informative, highlight the review concept and scope. As often abstract section is presented separately in search engines, it must be able to stand alone as an informative piece.

As suggested by the reviewer, we have introduced the indicated information (Lines: 15-17). However, we must point out that the abstract section is characterized by a limited number of characters.

Point 2: While the objective of this review is clear, authors should indicate also the review methodology. The consulted databases, keywords used to retrieve relevant literature studies, time spawn for the query.

As suggested by the reviewer, we have introduced the indicated information (Lines: 107-113).

Round 2

Reviewer 1 Report

I think that the authors' response is somewhat reasonable, so I would like to recommend this manuscript for publication.